# *TP53* Mutation in Acute Myeloid Leukemia: An Old Foe Revisited

**DOI:** 10.3390/cancers15194816

**Published:** 2023-09-30

**Authors:** Dong-Yeop Shin

**Affiliations:** 1Division of Hematology and Medical Oncology, Department of Internal Medicine, Seoul National University Hospital, Seoul 03080, Republic of Korea; shindongyeop@snu.ac.kr; Tel.: +82-2-2072-7209; Fax: +82-2-762-9662; 2Center for Medical Innovation, Biomedical Research Institute, Seoul National University Hospital, Seoul 03080, Republic of Korea; 3Cancer Research Institute, Seoul National University College of Medicine, Seoul 03080, Republic of Korea

**Keywords:** *TP53* mutation, acute myeloid leukemia

## Abstract

**Simple Summary:**

The *TP53* gene encodes the p53 protein, which plays a diverse role in responding to various cellular stresses. p53 consists of several functional domains, including a tetramerization domain for forming heterotetramers and a DNA-binding domain for bindings to p53-responsive elements. The most common mutations in acute myeloid leukemia (AML) occur in the DNA-binding domain of p53. *TP53* mutations are associated with a very poor prognosis and define a unique disease subgroup within AML. *TP53*-mutated AML often shows limited response to conventional chemotherapy and even allogeneic hematopoietic stem cell transplantation. There is an urgent need for novel treatment approaches for patients with *TP53*-mutated AML, given the frustrating treatment outcomes associated with this condition. Incorporating targeted agents and immunologic therapies into future treatment regimens may offer promising options for patients with *TP53*-mutated AML.

**Abstract:**

Introduction: *TP53* is the most commonly mutated gene in human cancers and was the first tumor suppressor gene to be discovered in the history of medical science. Mutations in the *TP53* gene occur at various genetic locations and exhibit significant heterogeneity among patients. Mutations occurring primarily within the DNA-binding domain of *TP53* result in the loss of the p53 protein’s DNA-binding capability. However, a complex phenotypic landscape often combines gain-of-function, dominant negative, or altered specificity features. This complexity poses a significant challenge in developing an effective treatment strategy, which eradicates *TP53*-mutated cancer clones. This review summarizes the current understanding of *TP53* mutations in AML and their implications. *TP53* mutation in AML: In patients with acute myeloid leukemia (AML), six hotspot mutations (R175H, G245S, R248Q/W, R249S, R273H/S, and R282W) within the DNA-binding domain are common. *TP53* mutations are frequently associated with a complex karyotype and subgroups of therapy-related or secondary AML. The presence of *TP53* mutation is considered as a poor prognostic factor. *TP53*-mutated AML is even classified as a distinct subgroup of AML by itself, as *TP53*-mutated AML exhibits a significantly distinct landscape in terms of co-mutation and gene expression profiles compared with wildtype (WT)-*TP53* AML. Clinical Implications: To better predict the prognosis in cancer patients with different *TP53* mutations, several predictive scoring systems have been proposed based on screening experiments, to assess the aggressiveness of *TP53*-mutated cancer cells. Among those scoring systems, a relative fitness score (RFS) could be applied to AML patients with *TP53* mutations in terms of overall survival (OS) and event-free survival (EFS). The current standard treatment, which includes cytotoxic chemotherapy and allogeneic hematopoietic stem cell transplantation, is largely ineffective for patients with *TP53*-mutated AML. Consequently, most patients with *TP53*-mutated AML succumb to leukemia within several months, despite active anticancer treatment. Decitabine, a hypomethylating agent, is known to be relatively effective in patients with AML. Numerous trials are ongoing to investigate the effects of novel drugs combined with hypomethylating agents, *TP53*-targeting agents or immunologic agents. Conclusions: Developing an effective treatment strategy for *TP53*-mutated AML through innovative and multidisciplinary research is an urgent task. Directly targeting mutated *TP53* holds promise as an approach to combating *TP53*-mutated AML, and recent developments in immunologic agents for AML offer hope in this field.

## 1. *TP53*—The First Tumor Suppressor Gene to Be Discovered

The human *TP53* gene located on chromosome 17p13.1 encodes p53 protein, a tumor suppressor protein which is 393 amino acids long and functions as a tetrameric transcription factor capable of regulating the expression of a huge set of target genes involved in the responses to cellular stress, including cell cycle arrest, DNA repair, apoptosis, and metabolism. *TP53* had once been thought to be an oncogene because mutant cDNA clones were capable of inducing cell transformation. Wild-type (WT) *TP53* was eventually classified as a tumor suppressor gene based on an experimental demonstration of its capacity for inhibiting the oncogenic transformation of cells in vitro [1]. *TP53* can be activated by various cellular stresses, including DNA damage, hypoxia, replicative stress, and oncogene expression. It primarily functions as a transcription factor, mediating various transcriptional regulations, depending on the type of cellular stress and cell type. For instance, minor DNA damage can result in cell cycle arrest and activate DNA repair mechanisms, whereas significant DNA damage may lead to senescence or apoptosis [2]. p53 is inactivated directly as a result of mutations in the *TP53* gene or indirectly as a result of alterations in genes whose products interact with p53 in about half of cancers. *TP53* is the most commonly mutated gene in human cancer cells; the amino-acid-changing mutation in the DNA-binding domain (DBD) of the *TP53* gene is frequently observed in various types of cancer, including colon, breast, lung, bladder, brain, pancreas, and stomach, as well as leukemia [3,4].

## 2. Structure and Physiologic Function of p53 Protein

The p53 protein is composed of an N-terminal transactivation motif, a DNA-binding domain, and a tetramerization domain (Figure 1). The transactivation domain interacts with the transcription machinery, whereas the tetramerization domain and the DNA-binding domain are involved in the formation of the p53 tetramer, which interacts with specific DNA target sequences called p53 response elements (p53 REs), comprising two copies of the 10-base-pair motif 5′-PuPuPuC(A/T)(T/A)GPyPyPy-3′ separated by 0–13 base pairs [5,6].

p53 dimers are generated co-translationally on the polyribosomes, and then two dimers gather and form a tetramer, which is an active form of p53 in cytosol or on DNA-binding sites [7]. p53 has more than 10 isoforms, including a canonical p53α (full-length p53), p53β, and p53γ, which can be translated by alternative splicing, or by the usage of an alternative promoter or alternative initiation of translation [8,9,10,11,12]. Each isoform possesses unique functional properties regarding interactions with other proteins. The diverse functions of various p53 isoforms can be implicated in various physiological roles in an organ-specific manner, thus giving rise to their pleiotropic biological activities from a single *TP53* gene. Each p53 isoform is regulated by small interfering RNA in an isoform-specific manner, providing the wide range of regulatory mechanisms of p53 [13]. Although p53-regulated genes may vary, depending on their isoforms, the fundamental characteristic of forming a p53 tetramer through the tetramerization domain remains largely consistent among different p53 isoforms. A switch for p53 signaling activation is normally ‘off’. However, post-translational modification, including phosphorylation and acetylation on the p53 protein induced by exogenous and endogenous stress like radiation and DNA damage can result in the decoupling of the p53 protein from the MDM2 protein, which leads to p53 activation [14]. As a result, the activated p53 protein can enter the nucleus, where it induces the expression of a plethora of target genes [15]. In response to cellular stress such as DNA damage, activated p53 proteins can induce cell cycle arrest or apoptosis of the respective cells by binding to genes such as *p21/WAF* and *Bcl-XL*. Aside from its role in cellular stress response, p53 plays a vital role in multiple metabolic regulations. p53 is involved in the synthesis and storage of fatty acids through the targeting of *SREBP1* and *ABCAs*, enhances gluconeogenesis and glycolysis by binding *SIRT6* and *GLUTs*, and regulates amino acid metabolism by targeting *SLC7As* and *PRODH* [16]. Furthermore, p53’s functional scope extends to epigenome regulation. In mouse embryonic stem cells, p53 represses DNA methylation by downregulating *DNMT3A* and *DNMT3B* [17]. Additionally, p53 actively participates in lineage commitment and cellular differentiation of stem cells through epigenetic regulation [18] (Figure 2).

## 3. Types of *TP53* Mutations and Their Effects on p53 Proteins

The most common type of *TP53* mutation in cancers is missense mutation in the DBD. There are six commonly detected mutations in the DBD (R175H, G245S, R248Q/W, R249S, R273H/S, and R282W), which are called the “six hotspot mutations”. Although these hotspot mutations are the most common, they comprise only around one-quarter of all *TP53* mutations in human cancers, because the *TP53* missense mutations have a remarkably broad spectrum and extreme diversity over various tumor types. A specific amino acid change due to mutations in the DBD of *TP53* causes the loss of DNA-binding potential, and *TP53* mutants at the six hotspots can be classified as a loss-of-function mutation. [19,20]. Cancer-associated *TP53* mutants can have the capability of abrogating WT *TP53*-induced transactivation, which is called “dominant-negative potential”, since the p53 heterotetramer consisting of the WT *TP53* dimer and *TP53* mutant dimer is generated in tumor cells harboring the *TP53* mutation and WT TP53 as heterozygotes [21]. Interestingly, some *TP53* mutants with the loss-of-function characteristic also have a gain-of-function activity. This feature was proved by a knock-in mouse model showing a more invasive and metastatic phenotype in *Trp53* mutants compared with *Trp53*−/− or *Trp53*+/− mice [22,23,24]. Some *TP53* mutations can exhibit different transactivating activities. The p.S121F *TP53* mutant is not capable of inducing the *CDKN1A* gene, but it is capable of transactivating other target genes such as *BAX, BBC3*, and *TNFRSF10B*, and inducing apoptosis [25,26]. Some mutations, including R282W, showed partial p53 functionality with or without temperature sensitivity, whereas some other mutations, including R337H, show wildtype-like activity or even super-transactivating activity. The knowledge about these variable genotype–phenotype correlations has been provided largely by dedicated experimental work by Ishioka’s group [27] and others [28,29,30]. Based on the effects of *TP53* mutations investigated in their work, *TP53* mutations can be classified as follows [31]: (1) loss-of-function mutation, (2) partial function with or without temperature sensitivity, (3) wild-type-like or super-transactivating, (4) with altered specificity (i.e., active or partially active on some targets but inactive on others), (5) dominant-negative mutation, and (6) gain-of-function mutation (acquisition of novel oncogenic activities, not shared with the WT protein). The most plausible analogy for this diversity of *TP53* mutations in terms of binding affinity toward variable target response elements might be a “hand” playing many keys with variable intensity on the piano, which was proposed by Resnick and Inga [29,32].

## 4. Scoring Systems of *TP53* Mutations for Clinical Application

There have been several suggestions regarding how to determine the impact of different *TP53* mutations on the clinical outcomes of patients with *TP53*-mutation-harbored cancers. Poeta et al. classified *TP53* mutation as disruptive and non-disruptive mutations, based on the location of the mutation and the predicted amino acid alterations [33]. They defined all DNA sequence alterations that introduce a STOP sequence resulting in disruption of p53 protein production or any DNA sequence alteration, which occurs within the L2 or L3 binding domains (codons 163–195 or 236–251) and replaces an amino acid from one polarity/charge category with an amino acid from another category as among four categories of nonpolar, negatively charged, polar with no charge, and positively charged. In this model, disruptive *TP53* mutation had clinical significance in terms of overall survival in patients with head and neck squamous cell carcinoma. Neskey et al. proposed an Evolutionary Action Score of *TP53* (EAp53) using a computational model based on the phenotype–genotype relationship of *TP53* mutations in the DBD. Scores ranging from 0 to 100 were calculated and outcomes of head and neck cancers with high and low EAp53 scores were compared [34]. These two models were established for predicting *TP53*-mutated head and neck cancers, and for consistent application to patients with AML. Eran Kotler et al. tested the functional impact of nearly every kind of *TP53* mutation in the DBD in a colon cancer cell line using a synthetically designed library. They performed extensive experiments to measure the proliferative capacity of each *TP53* variant relative to that of the WT p53-coding sequence (synonymous *TP53* mutation), and proposed a prediction model called “relative fitness score” [35]. In a recent investigation, these prediction scores for *TP53* mutations were applied to patients with AML [36]. In this study, the authors applied three different prediction models (disruptive vs. non-disruptive, EAp53, and RFS) to a cohort of patients with *TP53*-mutated AML. Variables of disruption (yes/no) and EAp53 (<75 or ≥75) did not prove to be significant predictive factors for OS and EFS. However, RFS (≤−0.135 or >−0.135) emerged as a significant predictive factor for OS (12.9 months for patients with RFS ≤ −0.135 and 5.5 months for patients with RFS > −0.135, *p* = 0.017) and EFS (7.3 months for patients with RFS ≤ −0.135 and 5.2 months for patients with RFS > −0.135, *p* = 0.033) in multivariate analysis. Although this last prediction model can be applied to patients with *TP53*-mutated AML, it was not derived from a tissue-specific approach targeting *TP53*-mutated cell types, but rather from a mutation-specific approach. Consequently, we anticipate future prediction models based on findings from *TP53* mutations in AML cells.

## 5. Evolution of *TP53*-Mutated Preleukemic HSPCs to AML

*TP53* mutations are found in less than 10% of patients with newly diagnosed AML; however, they increase remarkably by up to 50% or more in the setting of therapy-related AML [37]. *TP53* mutation can emerge after irradiation or cytotoxic stress [38]. A longitudinal study tracking the evolution of mutation has demonstrated that *TP53* mutations represent the primary mutational event in chemotherapy- or radiation therapy-induced MDS/AML. However, intriguingly, pre-existing *TP53* mutations exhibit resistance to chemotherapy or irradiation, and selectively expand after treatment. It is not the case that cytotoxic stress does not directly induce *TP53* mutations [39]. Healthy people with *TP53*-mutated hematopoietic clones have the highest probability of developing AML within a median of 4.9 years following the detection of *TP53* mutation in the blood [40]. Even though *TP53*-mutated HSCs represent only a small fraction of the total hematopoietic stem/progenitor pool, they keep expanding upon their survival advantage and preferentially proliferate through selective survival pressure during chemotherapy and allogeneic hematopoietic stem cell transplantation [41]. Although *TP53* is one of the genes found in age-related clonal hematopoiesis [42], it appears that *TP53* exerts a more deterministic influence compared with other genes such as *DNMT3A, TET2*, and *ASXL1* when it comes to leukemic transformation. Further, with aging, HSCs gradually lose their self-renewal capacity and shift toward myeloid lineage with decreases in lymphopoiesis [43] and erythropoiesis [44]. The cell of origin of AML can be an HSC or a myeloid progenitor cell such as an MPP, MEP, or GMP. It is worth noting that terminally differentiated hematopoietic cells like granulocytes may have limited opportunities to preserve acquired somatic mutations, due to their relatively short lifespan. Some mutations, such as *DNMT3A, TET2, RUNX1*, and *EZH2*, are suggested to have originated in HSCs, based on observations that these mutations have been detected in T-cells in patients with AML, which was called “lympho-myeloid clonal hematopoiesis” [45,46]. In the case of *TP53*-mutated leukemia, both HSCs and HPCs are thought to be cellular origins [46]. *TP53* mutation has been associated with LMPP-like LSC dominant phenotype in patients with newly diagnosed AML in terms of LSC immunophenotype [47]. Additional mutations are accumulated in preleukemic clones with *TP53* mutation over time, and these preleukemic HSCs with multiple mutations co-exist with leukemic bulk at initial diagnosis, persist during intensive chemotherapy, and become sources of relapse in some cases [48,49,50]. Currently, a stepwise leukemogenesis model indicating that a preleukemic state with founder mutation of epigenetic regulator genes like DTA (*DNMT3A, TET2*, and *ASXL1*) preludes fully transformed leukemic cells, with additional mutation on signaling or transcriptional genes like *RAS, FLT3, NPM1*, or *CEBPA*, is widely accepted [51]. However, *TP53* mutation is enriched in patients with hematologic malignancies without DTA (*DNMT3A*, *TET2*, and *ASXL1*) mutation [52], which suggests that leukemogenesis by *TP53* mutation is somewhat different from that caused by DTA mutations. Rather, *TP53* mutations commonly co-occur with cytogenetic abnormalities, including deletion of chromosomes 5, 7, and 17 or complex abnormalities of >3 [53,54].The leukemogenesis mechanism in *TP53*-mutated AML might involve multiple hits accelerated by a deficient DNA repair mechanism resulting from *TP53* mutation rather than epigenetic dysregulation due to DTA mutations. Due to this distinct genetic feature, one of the recent classification systems classifies TP53-mutated AML as a separate disease entity [55]. Transformation from preleukemic stem cells to leukemic stem cells has repeatedly been shown to accompany an increasing number of gene mutations. However, the question regarding which factors critically contribute to leukemic transformation from a preleukemic state is still unsolved.

## 6. Genetic Characteristics of *TP53*-Mutated AML

Some co-occurring mutations are detected in patients with *TP53*-mutated AML. Epigenetic genes such as *DNMT3A, ASXL1*, and *TET2* or RAS/MAPK signaling genes including *NF1, KRAS/NRAS*, and *PTPN11* or transcription factors like *RUNX1* or genes involving RNA splicing such as *SRSF2* are frequently mutated in the same leukemic clone or sub-clones. *SRSF2*, *RUNX1* and *ASXL1* frequently occur in patients with newly diagnosed AML with one *TP53* mutation, whereas *KRAS/NRAS, PTPN11*, and *RUNX1* commonly occur in ≥2 *TP53* mutations [56]. Another study comparing co-occurring mutations in *TP53*-mutated AML and *TP53*-WT AML demonstrated that *FLT3, NPM1,* and *RAS* gene mutations are not common in *TP53*-mutated AML compared with *TP53*-WT AML, whereas other gene mutations are not different between the two groups [57]. However, *TP53* mutation is also associated with chromosome-level alterations such as complex karyotype, aneuploidy including del(5q), −5, −7, and del (3p) rather than single nucleotide variation. The frequency of *TP53* mutation in MDS/AML with a complex karyotype is reported to be up to 83% [54]. The high incidence of *TP53* mutation in MDS/AML is closely linked to the presence of a complex karyotype. Both of these abnormalities are associated with prior therapeutic interventions that induce DNA damage. However, it remains unclear how *TP53* mutation contributes to the development of a complex karyotype in hematopoietic stem and progenitor cells. Chromothripsis events are thought to be a cause of complex karyotypes in *TP53*-mutated AML [58]. There are increased telomere contents in *TP53*-mutated AML compared with other AMLs [59]. Several studies have investigated the characteristics of the gene expression profile of *TP53*-mutated AML. Immune infiltration in the bone marrow microenvironment is characteristic of *TP53*-mutated AML [60]. IFN-α and IFN-γ signaling of the T-cells in the peripheral blood of patients with *TP53*-mutated AML are stronger than in healthy donors [61]. Increased immune cell infiltration in *TP53*-mutated AML might be partly a result of increased tumor mutation burden, considering the observation of increased tumor mutation burden in various cancer types with *TP53* mutation [62]. A previous study quantitating the infiltration of T-cells and their receptors using immunohistochemical staining on bone marrow cells showed increased PD-L1 expression in *TP53*-mutated AML cells [63]. However, the addition of PD-L1 inhibitor or PD-1 inhibitor on azacytidine did not increase the response rate in patients with *TP53*-mutated AML at all [64,65]. Nevertheless, other immunologic agents including CD123-targeting bispecific DART or CD47-targeting monoclonal antibodies showed promising results, providing hope to patients with *TP53*-mutated AML [60,66] (Table 1).

## 7. Despair and Hope in *TP53*-Mutated AML: Is the Future Bright?

*TP53*-mutated AML is a clinically devastating disease, which responds poorly to standard therapy including allogeneic hematopoietic stem cell transplantation. The rate of complete remission after the first induction chemotherapy in *TP53*-mutated AML is not only decreased to around 50%, but it is also observed that remission is not durable. As a result, median overall survival in *TP53*-mutated AML is from 2 to 10 months [83,84]. *TP53*-mutated AML is now classified as a separate disease entity from other subtypes of AML [85]. *TP53*-mutated AML is highly associated with a complex karyotype, a very poor prognostic factor. The frequency of *TP53* alteration including *TP53* mutation and loss in AML with a complex karyotype is up to 70% [86]. Low *TP53* mutation burden (only 1 *TP53* mutation and VAF ≤ 40%) is associated with better survival in *TP53*-mutated AML [87]. A standard chemotherapy regimen including high-dose cytarabine increases OS only in patients with a low burden of *TP53* mutation. In contrast, treatment outcome is independent of mutated *TP53* VAF when treated with a prolonged schedule (10 days) of decitabine [87]. A cornerstone trial of decitabine for adult patients with AML/MDS demonstrated a surprising result: patients with *TP53* mutation showed a remarkably higher response rate than patients without *TP53* mutation (100% (21/21) vs. 41% (32/78), *p* < 0.001) [67]. Furthermore, *TP53*-mutated leukemic clones disappeared as a result of decitabine treatment in many cases. It remains unclear why hypomethylating agents are effective in *TP53*-mutated AML. However, a plausible explanation for this phenomenon is that hypomethylating agents like decitabine may reverse an unbalanced DNA methylation in *TP53*-mutated AML cells because normal p53 suppresses DNA methylation by up-regulating *TET1/2* components of the demethylation machinery [17]. Following this discovery, several studies have tested whether the addition of other agents to hypomethylating agents is synergistic in *TP53*-mutated AML. However, this unrealistic CR rate was not represented in other studies where a combination with 10-day decitabine improved the outcomes of *TP53*-mutated AML. In a randomized phase II study to investigate the addition of bortezomib, a proteasome inhibitor on 10-day decitabine showed a poor response in patients with *TP53*-mutated AML [69]. A substudy of the Beat AML Master Trial tested the combination of a Syk inhibitor, entospletinib, and 10-day decitabine as a phase II study, which also demonstrated a disappointingly low CR rate of 13.3% and short median OS of 6.5 months [70] (Table 1). However, the addition of venetoclax, a promising novel drug in the field of AML treatment, did not increase the response rate of *TP53*-mutated AML. Two studies investigated the benefits of venetoclax in patients with *TP53*-mutated AML. In the first study, a prospective phase II trial, investigators tested 10-day decitabine plus venetoclax 400 mg daily followed by 5-day decitabine plus venetoclax after achieving remission in newly diagnosed AML. The overall response rate and complete remission rate were 89% and 77%, respectively, in WT-*TP53* AML, whereas they were 66% and 57%, respectively, in *TP53*-mutated AML. Survival outcomes were less favorable regarding response rate, with a 60-day mortality of 26% vs. 4% and overall survival of 5.2 and 19.4 months in *TP53*-mutated and WT-*TP53* AML, respectively [68]. In the second study, a retrospective analysis aimed to compare venetoclax-based and non-venetoclax-based regimens in *TP53*-mutated AML. In this study, investigators could not identify a significant difference between the venetoclax-based regimen and the non-venetoclax-based regimen in terms of OS (median OS, 6.6 vs. 5.7 months, *p* = 0.4) and relapse-free survival (median relapse-free survival, 4.7 vs. 3.5 months, *p* = 0.43) [88]. The first study was prospective and had a homogeneous population with the same treatment protocol, but it lacked a comparator group without venetoclax. On the other hand, the second study compared *TP53*-mutated and *TP53*-nonmutated AML, despite treatment being heterogeneous and retrospective.

Azacytidine, another hypomethylating agent, is widely used as a partner for venetoclax in the treatment of elderly patients with AML. A cornerstone phase III study was conducted to confirm the superiority of venetoclax plus azacytidine compared with azacytidine alone [74]. However, the efficacy of azacytidine plus venetoclax in the subgroup of patients with *TP53* mutation was unsatisfactory (Table 1). Pevonedistat, a NEDD8-activating enzyme inhibitor, was one of the candidates for a synergistic combination with hypomethylating agents. However, a phase II study to investigate the efficacy of pevonedistat and azacytidine, which was initiated based on a preliminary observation of activity for *TP53*-mutated AML in a phase I study, was terminated early, due to no efficacy [75].

Considering data obtained so far from drugs which are currently clinically available or soon to be available, there are two paths worth placing our hopes in. The first one is mutated-p53-targeted therapy, which is called p53 reactivators. APR-246 (eprenetapopt) is a prodrug converting to MQ, which binds to a specific thiol group in the p53 DNA-binding domain. Mutant p53 can be refolded to its wildtype configuration and reactivated through MQ binding [89,90]. In this way, APR-246 targets the mutated p53 and reactivates p53 activity. APR-246 showed a potential activity in *TP53*-mutated MDS/AML in a phase Ib/II study (ORR 73% and CR 50% in *TP53*-mutated MDS and ORR 64% and CR 36% in *TP53*-mutated AML) [77] and another phase II study (ORR 62% and CR 47% in *TP53*-mutated MDS and ORR 33% and CR 17% in *TP53*-mutated AML) [78]. Doublet therapy of eprenetapopt and azacytidine was tested as a maintenance therapy for one year after allogeneic hematopoietic stem cell transplantation in *TP53*-mutated AML/MDS. In this study, median relapse-free survival and overall survival were 12.5 and 20.6 months, respectively [79]. This result is a promising one, considering the dismal outcome of *TP53*-mutated MDS/AML. APR-246 was also tested as a triple combination therapy with azacytidine and venetoclax in *TP53*-mutated AML. Complete remission and CR/CRi/MLFS rate in this phase I study were 38% and 59%, respectively. This result indirectly supports the previous finding that the addition of venetoclax is not beneficial in *TP53*-mutated AML, though we consider that an interpretation of treatment outcomes from a phase I study should be cautious [80]. COTI-2, another small molecule, also has the capability to restore DNA binding of p53 by inducing a conformational change of mutated p53 toward wildtype p53 [91]. Arsenic trioxide, a standard treatment agent for acute promyelocytic leukemia, is also known to rescue structural mutated p53 by stabilizing the DNA-binding area. Arsenic trioxide may be a candidate drug for repurposing, to target *TP53*-mutated AML [92]. However, these molecules have not yet been tested on human subjects with *TP53*-mutated malignancy in a clinical trial context. Several methods to indirectly target mutated *TP53* beyond *TP53* reactivators have also been studied in *TP53*-mutated solid cancers. One method constitutes the enhanced degradation of mutated p53 by protein degraders, including heat shock protein 90 inhibitor [93], whereas others use the strategy of synthetic lethality by blocking G2/M or S-phase cellular arrests using ATR/CHK1/WEE1 inhibitors [94,95]. These approaches will also be worth exploring in patients with *TP53*-mutated AML in the near future.

The second method uses immunologic agents. There is a suggestion that *TP53*-mutated AML could be effectively treated using novel agents with immunologic mechanisms. Flotetuzumab, a CD123 x CD3 bispecific dual-affinity retargeting antibody (DART) molecule, demonstrated a promising efficacy in relapsed/refractory *TP53*-mutated AML, showing a 47% complete remission rate and a 10.3-month OS [60]. Magrolimab, a monoclonal antibody targeting CD47, an interesting signaling pathway which is interpreted as “Don’t eat me” by tumor-infiltrating macrophages, has showed potential activity for effectively treating *TP53*-mutated AML. In a phase I/II study, the CR/CRi rate in patients with *TP53*-mutated AML was 100% (7/7) [66]. Several trials to investigate the efficacy of magrolimab combination therapies with azacytidine and venetoclax are currently ongoing. Another piece of supporting evidence regarding the treatment efficacy of *TP53*-mutated AML by immunologic mechanism comes from the current clinical practice. A recent retrospective study extracted independent predictive factors for prolonged OS in a cohort of patients with *TP53*-mutated AML treated with allogeneic stem cell transplantation [96]. In this study, only two clinical factors of CR/CRi on day 100 post allogeneic HSCT and chronic GVHD could predict prolonged OS, which indirectly supports the clinical significance of the role of immunologic mechanisms through a graft-versus-leukemia effect. Another retrospective study from the MD Anderson Cancer Center showed a prolonged OS in *TP53*-mutated AML treated with allogeneic HSCT compared with *TP53*-mutated AML not treated with allogeneic HSCT (median OS 33.7 months vs. 7.0 months), but the Kaplan–Meyer graph in this article strongly suggested that even allogeneic HSCT is not curative for *TP53*-mutated AML [97]. Another promising approach is the recently proposed concept of immunological targeting of *TP53*-mutation. A certain hotspot for *TP53* mutation is known to be the immune-cell-stimulating neoantigen, and this can be effectively targeted by T-cells engaging bispecific antibodies [98]. Although this approach is still in a very early stage of research and development, it should be actively studied considering its potential for the cure of *TP53*-mutated AML.

## 8. Conclusions

*TP53*-mutated AML is a unique subtype of AML with a very poor prognosis. *TP53*-mutated AML is highly enriched in therapy-related or secondary AML with a complex karyotype. This type of *TP53* mutation is heterogeneous, but the most common types of *TP53* mutation in AML are mainly missense mutations in the DNA-binding domain, which are not only characterized by loss of function in view of loss of effective DNA sensing of the target sequence and resultant physiologic regulatory function and/or dominant-negative effect, but are also often characterized by gain of function in terms of gain of mutated oncogenic p53 proteins contributing to the malignant phenotype. The current standard treatment strategies for AML make it difficult to induce durable remission and to cure *TP53*-muated AML. Five-day or ten-day decitabine could be the preferred options for *TP53*-mutated AML. Standard therapy including high-dose cytarabine could be considered as an initial induction therapy for some patients with treatment-naive *TP53*-mutated AML with a low *TP53* mutation burden. Allogeneic HCT should be considered in transplant-eligible patients with *TP53*-mutated AML, despite the low probability of long-term survival. The incorporation of *TP53* reactivators and immunologic agents including CD123-targeted DART and CD47-targeting immune-checkpoint inhibitor is now being prepared for running, at the starting line, after completing early-phase clinical trials.

## Figures and Tables

**Figure 1 cancers-15-04816-f001:**
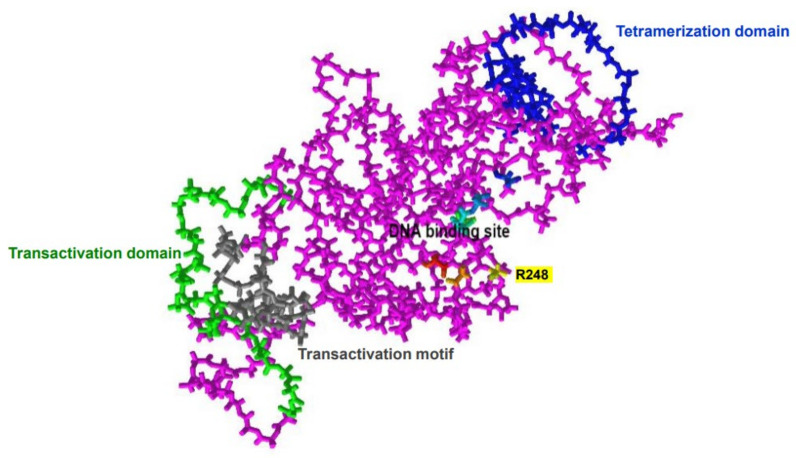
Human p53 protein structure (monomer). p53 consists of a transactivation motif (Residue 6–30, gray), transactivation domain (35–59, lime), DNA-binding domain (109–288), and tetramerization domain (319–355, blue). The DNA-binding site contains R248, which is most commonly mutated in AML. Available online: https://www.ncbi.nlm.nih.gov/Structure/pdb/8F2I (accessed on 30 July 2023).

**Figure 2 cancers-15-04816-f002:**
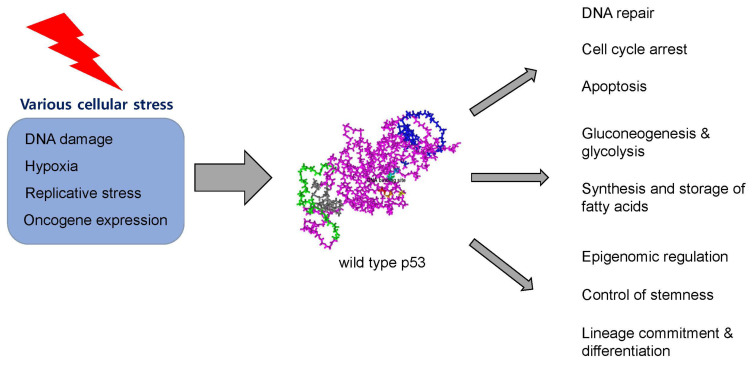
Function of p53. The wild-type p53 protein plays a diverse range of physiological roles, including involvement in DNA repair, cell cycle arrest, apoptosis, glucose and lipid metabolism, epigenomic regulation, control of stemness, lineage commitment, and differentiation. These functions are triggered in response to various cellular stresses, such as DNA damage, hypoxia, replicative stress, and oncogene expression.

**Table 1 cancers-15-04816-t001:** Published results of clinical trials for *TP53*-mutated AML.

Drug	Phase	Study Population	Treatment and Mechanism of Action	*TP53*-Mutated AML Sub-Population	Response	Survival	Trial ID	Ref.
Hypomethylating Agents and Their Combinations								
Decitabine	II	AML/MDS with adverse-risk karyotype (n = 116)	10-day decitabine (Hypomethylating agent)	10% (12 *TP53*-mutated AML, 9 *TP53*-mutated MDS)	CR/CRi/MLFS 46% in all MDS/AML, CR/CRi/MLFS 100% (21/21) in *TP53*-mutated MDS/AML	mOS 12.7 months in *TP53*-mutated MDS/AML, mOS 15.4 months in *TP53*-WT MDS/AML	NCT01687400	[67]
Decitabine + venetoclax	II	Newly diagnosed AML (n = 118)	10-day decitabine + venetoclax (BCL-2 inhibitor)	30% (35 *TP53*-mutated AML)	ORR 66% and CR/CRi 57% in mTP53 AML, ORR 89% and CR/CRi 77% in *TP53*-WT AML	mOS 5.2 months in *TP53*-mutated AML, 19.4 months in *TP53*-WT AML	NCT03404193	[68]
Decitabine ± bortezomib	II, randomized	Newly diagnosed elderly AML (n = 163)	10-day decitabine vs. decitabine + bortezomib (proteasome inhibitor)	22%	CR 21% in TP53-mutated AML treated with decitabine, CR 17% in *TP53*-mutated AML treated with decitabine plus bortezomib	1 yr OS 18% in *TP53*-mutated AML, 51% in *TP53*-WT AML	NCT014230926	[69]
Decitabine + entospletinib	II	Newly diagnosed elderly AML with *TP53* mutation (cohort A, n = 45) or complex karyotype (cohort B, n = 13)	10-day decitabine + entospletinib (Syk inhibitor)	78% (45/58)	CR/CRi and CR/CRi/MLFS 33.3% and 48.9% in *TP53*-mutated AML, CR/CRi and CR/CRi/MLFS 61.5% and 76.9% in *TP53*-WT and complex karyotype	mOS 6.5 months in *TP53*-mutated AML, mOS 11.5 months in *TP53*-WT and complex karyotype	NCT03013998	[70]
Decitabine ± ibrutinib	II, randomized	Untreated AML (n = 144)	10-day decitabine (n = 72) vs. decitabine + ibrutinib (n = 72) (BTK inhibitor)	19% (27/144)	CR/CRi 56% in *TP53*-mutated AML treated with decitabine +- ibrutinib	mOS about 6 months in *TP53*-mutated AML, mOS about 12 months in *TP53*-WT AML	EudraCT 2015-002855-85	[71]
Decitabine + cladribine/LDAC	II	Untreated elderly AML	5-decitabine alternating with cladribine and LDAC	17% (20/118)	CR/CRi 40%	mOS 5.4 months	NCT01515527	[72]
Azacytidine	Prospective observational	MDS/AML/CMMoL (n = 62)	Azacytidine (hypomethylating agent)	15% (9 *TP53*-mutated AML, 14 *TP53*-mutated MDS, 39 *TP53*-WT)	CR 22% and ORR 44% in *TP53*-mutated MDS/AML CR 38% and ORR 51% in *TP53*-WT MDS/AML	mOS 12.4 months in *TP53*-mutated AML/MDS, 23.7 months in *TP53*-WT AML/MDS	N/A	[73]
Azacytidine ± venetoclax	III	Untreated AML (n = 431)	Azacytidine + venetoclax vs. azacytidine	12% (52 *TP53*-mutated AML, 379 *TP53*-WT AML)	CR/CRi 40.8% and ORR 55.3% in *TP53*-mutated AML with poor cytogenetics treated with azacytidine and venetoclax	mOS 5.17 and 4.9 months in *TP53*-mutated AML with poor-risk cytogenetics treated with azacytidine ± venetoclax	NCT02993523	[74]
Azacytidine + pevonedistat	II	*TP53*-mutated AML (n = 10)	Azacytidine + pevonedistat (NEDD8-activating enzyme inhibitor)	100% (10/10)	No CR, 2 PR, 7 SD	mOS 6.3 months	NCT03013998	[75]
***TP53*-targeting agents and their combinations**								
Idasanutlin ± cytarabine	I	R/R AML (n = 122)	Idasanutlin (MDM2 inhibitor) (n = 46), idasanutlin + cytarabine (n = 76)	20% (25/122)	CRc = 18.9% and 35.6% in idasanutlin and + cytarabine. CRc = 4.0% (1/25) in *TP53*-mutated AML	DoR = 7.7 and 8.5 months in idasanutlin and + cytarabine arm in *TP53*-WT AML (data of *TP53*-mutated AML are N/A)	NCT017773408	[76]
APR-246 + azacytidine	Ib/II	HMA-naive MDS/AML (n = 55)	APR-246 (*TP53* reactivator) + azacytidine	20% (11 AML, 40 MDS 4 MDS/MPN)	ORR 64% CR 36% in AML, ORR 73% and CR 50% in MDS	mOS 10.8 months for AML, 10.4 months for MDS	NCT 03072043	[77]
APR- 246 + azacytidine	II	Treatment-naive *TP53*-mutated MDS/AML (n = 52)	APR-246 + azacytidine	35% (18 AML, 34 MDS)	CR/CRi (47% in MDS, 45% in 20–30% blast AML, 14% in >30% blast AML)	mOS 12.1 months for MDS, 10.4 months for AML	NCT 03072043	[78]
APR-246 + azacytidine	II	*TP53*-mutated MDS/AML receiving allogeneic HCT (n = 33)	APR + azacytidine maintenance for 12 cycles	42% (14 AML, 19 MDS)		mRFS 12.5 months mOS 20.6 months (for MDS/AML)	NCT 03931291	[79]
APR-246 + azacytidine + venetoclax	I	*TP53*-mutated AML (n = 49; APR-246 + venetoclax 6, APR-246 + venetoclax + azacytidine n = 43)	APR-246 + venetoclax + azacytidine	100% (de novo 45%, secondary 55%)	CR/CRi/MLFS 59%	Duration of CR 4.9 months mOS 7.3 months mOS (bridged to allogeneic HCT) 20.0 months	NCT 04214860	[80]
Azacytidine ± APR-246	III	*TP53*-mutated MDS/AML (n = 154)	Azacytidine vs. azacytidine + APR-246	N/A	CR 33.3% in azacytidine + APR-246 vs. CR 22.4% in azacytidine (*p* = 0.13)	N/A	NCT 03745716	Unpublished (press release only)
**Immunologic agents and their combinations**								
Ipilimumab + decitabine	I	R/R MDS/AML, (n = 54; 23 AML, 2 MDS, post-transplant (arm A) + 15 AML, 8 MDS, transplant-naive (arm B) + 6 N/E pts)	Ipilimumab (anti-CTLA-4 moAb) + Decitabine	22.2% *TP53*-mutated MDS/AML (12/54 including 8 pts without NGS data)	ORR 20% in post-transplant MDS/AML, 52.1% in transplant-naive pts (Data in *TP53*-mutated subgroup are N/A)	DOR = 4.46 and 6.14 months in with and without prior HSCT (data in *TP53*-mutated subgroup are N/A)	NCT02890329	[81,82]
Nivolumab + azacytidine	II	R/R AML (n = 70)	Nivolumab (anti-PD-1 MoAb) + azacytidine	23% (16/70)	ORR 18.8% in *TP53*-mutated AML, ORR 33% in R/R AML (all patients)	mOS 5.98 months in *TP53*-mutated AML, mOS 6.3 months in R/R AML (all patients)	NCT02397720	[64]
Durvalumab + azacytidine	II	Untreated MDS/AML (n = 213)	Durvalumab (anti-PD-L1 MoAb) + azacytidine	27.7% (59/213), 45.7% among AML (59/129)	CR/CRi 20% in aza/durvalumab arm, CR/CRi 23% in azacytidine arm ORR 35% in *TP53*-mutated AML, ORR 34% in *TP53*-WT AML	mOS and RFS 13.0 and 9.5 months in aza/durvalumab arm, mOS and RFS = 14.4 and 12.2 in azacytidine alone arm	NCT02775903	[65]
Flotetuzumab	I/II	R/R AML/MDS	Flotetuzumab (Anti-CD3/CD123 bispecific DART)	33% (15/45 R/R AML)	CR 47% (7/15) in *TP53*-mutated AML	mOS 10.3 months in *TP53*-mutated AML	NCT02152956	[60]
Magrolimab + azacytidine	Ib	Untreated, venetoclax-naive R/R or -exposed R/R	Magrolimab (anti-CD47 MoAb) + azacytidine		ORR 63% (27/43) in de novo AML ORR 69% (20/29) in *TP53*-mutated AML	mOS 12.9 months for *TP53*-mutated AML mOS 18.9 months for *TP53* WT AML	NCT04435691	ASH 2021 abstract

AML, acute myeloid leukemia; CRc, composite complete remission; CRi, complete remission with incomplete recovery; DART, dual affinity retargeting molecule; DoR, duration of response; MDS, myelodysplastic syndrome; MLFS, morphologic leukemia-free state; MoAb, monoclonal antibody; mOS, median overall survival; mRFS, median relapse-free survival; NGS, next-generation sequencing; ORR, overall response rate; PR, partial remission; R/R, relapsed or refractory; SD, stable disease; WT, wildtype.

## Data Availability

All data are included in this article.

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
