# Peer review of "TP53 Mutation in Acute Myeloid Leukemia: An Old Foe Revisited"

_cancers, 2023, doi:10.3390/cancers15194816_

Round 1
Reviewer 1 Report
This review aims to address important questions concerning the role of mutant p53 in AML, the potential mechanisms through which mutant p53 influences the AML phenotype, and strategies for therapeutically targeting AML with mutant p53. However, the resulting manuscript falls short of the publication standards expected by Cancers. The background information on p53 biology is cursory and, in some instances, inaccurate (e.g., the statement suggesting that p53 induces apoptosis depending on the severity of damage is wrong). Similar issues are observed in the sections describing the properties of p53 mutants and the drugs developed for targeting p53. The manuscript lacks information about the epigenetic repressor functions of p53, which could provide a plausible explanation for the high efficacy of DNA demethylating agents in AML with mutant p53. Furthermore, there is no thorough analysis of clinical trial results. Lastly, but not least, the authors' use of the English language requires significant improvement. In its current state, this manuscript falls short of fulfilling the authors' objectives and necessitates substantial revision before it can be considered for publication.
Provided in the main review section.
Author Response
-> Thank you for your invaluable advice. The primary objective of this review article is to explore potential treatment strategies by focusing on currently available and investigational anti-leukemic agents. The sections of p53 biology serve an introductory role in this article. I have revised the sentence you pointed out as inaccurate to read as follows.
“In response to cellular stress such as DNA damage, activated p53 proteins can induce cell cycle arrest or apoptosis in the respective cells by binding to genes such as p21/WAF and Bcl-XL.” I have also sequentially added information about the role of p53 in regulating metabolism and stemness. “Aside from its role in cellular stress response, p53 plays a vital role in multiple metabolic regulations. p53 is involved in the synthesis and storage of fatty acids through the targeting of SREBP1 and ABCAs, enhances gluconeogenesis and glycolysis by binding SIRT6 and GLUTs, and regulates amino acid metabolism by targeting SLC7As and PRODH. (16)” [Section 2, Page 3]
I highly appreciate your insightful comments regarding the role of p53 in the epigenome. Sentences describing the connection between p53 and DNA methylation have been added to the final paragraphs of section 2 and 7, as follows.
“Furthermore, p53’s functional scope extends to epigenome regulation. In mouse embryonic stem cells, p53 represses DNA methylation by down-regulating DNMT3A and DNMT3B. (17) Additionally, p53 actively participates in lineage commitment and cellular differentiation of stem cells through epigenetic regulation. (18)” [Section 2]
“It remains unclear why hypomethylating agents are effective in TP53-mutated AML. However, a plausible explanation for this phenomenon is that hypomethylating agents like decitabine may reverse an unbalanced DNA methylation in TP53-mutated AML cells because normal p53 suppresses DNA methylation by up-regulating TET1/2 components of the demethylation machinery. (17)” [Section 7]
Regarding clinical trials, I have provided a summary of trials in which patients with TP53-mutated AML were enrolled, and data on the TP53-mutated AML subgroup were available. Furthermore, I have excluded trials using outdated cytotoxic regimens from Table I, as this summarized table aims to provide insights into potential future anti-cancer drugs.
According to your request, I sought professional English editing services to further enhance the manuscript’s language quality. A certification issued by the MDPI author service is attached.

Reviewer 2 Report
TP53 mutations are commonly found in many human cancers and this review focusses on role of TP53 mutations in AML. The review is written in great details and the overall layout of the review article is well thought out. My only problem with the article is that parts of it are written as separate lines just pasted together. Overall flow needs to be worked on for many sections of this article.
Here are my comments and thoughts about the review:
1. Introduction of relative fitness score (RFS) in abstract is abrupt and not needed. No other scoring system is discussed in the abstract, then why is there mention of RFS?
2. The last line of abstract should have ‘mutations’ as there could be plenty of TP53 mutations.
3. There could be more detail about the discovery of TP53 as a tumor suppressor gene more detailed dive into the literature, just staing the fact that it a tumor suppressor gene and not an oncogene is not sufficient. What makes it a tumor suppressor? How is response to cellular stress through TP53 is linked to tumor suppression?
4. Domain word duplicated in section2.
5. While describing isoforms of p53, it might be interesting to discuss why would a protein like p53 have so many isoforms? What is the advantage to cells by making all of the different isoforms?
6. The discussion about the physiological functions of p53 is insufficient. There is no detailed discussion about its targets, how p53 affects these targets and how the regulation is different in different situations.
7. In section 3, DBD abbreviation is used first time without mention of its full form.
8. In section 4, authors change the method of writing and discuss one paper at a time explaining their findings and contribution. It would be better if this part is written as a coherent discussion instead. Authors could add their opinions, predictions, and speculations in a better way if this part is like a discussion.
9. In section 5 again, there could be more addition to the discussion rather than mere stating facts.
10. In the last paragraph of section5, there are no references. Are these all authors’ original ideas?
11. In section 6, authors discuss reports of 83% of TP53 mutation frequency in AML patients without discussing it further. It might be good to discuss if this is higher frequency than predicted? If so, why would it be higher? What roles of TP53 would be important for such a phenotype?
12. Right after talking about mutation frequency and complex karyotype, authors jump to chromothrypsis and telomere content very abruptly. Some introduction about chromothrypsis particularly why chromathrypsis may be connected to TP53 mutations, complex karyotypes etc. might be useful for readers. Similarly, before jumping to telomeres, why TP53 may be important for telomere maintenance could add some context for the readers.
13. Paragraph about single-institute trail of decitabine could be re-written with better flow and coherence.
14. Instead of writing ‘this result was also confirmed by a retrospective study’ authors could discuss the two studies together and state merits or demerits of one study over the other.
15. The discussion about ongoing clinical trails for TP53 mutated AML patients is well-written and the idea of having both table and discussion is well suited for this review’s purpose.
The quality of English is good. There are some mistakes that can be easily fixed.
Author Response
TP53 mutations are commonly found in many human cancers and this review focusses on role of TP53 mutations in AML. The review is written in great details and the overall layout of the review article is well thought out. My only problem with the article is that parts of it are written as separate lines just pasted together. Overall flow needs to be worked on for many sections of this article.
-> Thank you for your comments. As you pointed out, there were many sections in my original manuscript where sentences were written as separate lines. I have now removed those extra spaces between separate sentences in most parts.
Here are my comments and thoughts about the review:
- Introduction of relative fitness score (RFS) in abstract is abrupt and not needed. No other scoring system is discussed in the abstract, then why is there mention of RFS?
-> I mentioned of RFS in the abstract because I believed it to be one of the most crucial tools for predicting the prognostic implications of different TP53 mutations in AML. However, it appeared somewhat abrupt in the context of the abstract. Therefore, I have revised removed the following sentence “Among those scoring systems, relative fitness score (RFS) could be applied to AML patients with TP53 mutations.”
- The last line of abstract should have ‘mutations’ as there could be plenty of TP53 mutations.
-> Revised as suggested.
- There could be more detail about the discovery of TP53 as a tumor suppressor gene more detailed dive into the literature, just staing the fact that it a tumor suppressor gene and not an oncogene is not sufficient. What makes it a tumor suppressor? How is response to cellular stress through TP53 is linked to tumor suppression?
-> The transcriptional regulation of TP53 as a tumor suppressor was described as follows in section 1.
“TP53 can be activated by various cellular stresses, including DNA damage, hypoxia, replicative stress, and oncogene expression, It primarily functions as a transcription factor, mediating various transcriptional regulations depending on the type of cellular stress and cell type. For instance, minor DNA damage can result in cell cycle arrest and activate DNA repair mechanisms, while significant DNA damage may lead to senescence or apoptosis.(2)”
- Domain word duplicated in section2.
-> Thank you for your comment. Duplicated domains have been removed.
- While describing isoforms of p53, it might be interesting to discuss why would a protein like p53 have so many isoforms? What is the advantage to cells by making all of the different isoforms?
-> You’re correct. The production of multiple isoforms can yield a broad spectrum of transcriptional regulation from a single TP53 gene. The following paragraph has been added in section 2.
“Each isoform possesses unique functional properties regarding interactions with other proteins. The diverse functions of various p53 isoforms can be implicated in various physiological roles in an organ-specific manner, thus giving rise to their pleiotropic biological activities from a single TP53 gene. Each p53 isoform is regulated by small interfering RNA in an isoform-specific manner, providing a wide range of regulatory mechanisms for p53. (13) Although p53-regulated genes may vary depending on its isoforms, the fundamental characteristic of forming a p53 tetramer through OD remains largely consistent among different p53 isoforms.”
- The discussion about the physiological functions of p53 is insufficient. There is no detailed discussion about its targets, how p53 affects these targets and how the regulation is different in different situations.
-> Physiological functions of p53, including cellular stress response and others, have been added. Examples of p53 targets in different situations have also been included, as follows.
“In response to cellular stress such as DNA damage, activated p53 proteins can induce cell cycle arrest or apoptosis in the respective cells by binding to genes such as p21/WAF and Bcl-XL.” Additionally, information about the role of p53 in regulating metabolism and stemness has been sequentially added. “Aside from its role in cellular stress response, p53 plays a vital role in multiple metabolic regulations. p53 is involved in the synthesis and storage of fatty acids through targeting of SREBP1 and ABCAs, enhances gluconeogenesis and glycolysis by binding SIRT6 and GLUTs, and regulates amino acid metabolism by targeting SLC7As and PRODH. (16) Furthermore, p53’s functional scope extends to epigenome regulation. In mouse embryonic stem cells, p53 represses DNA methylation by down regulating DNMT3A and DNMT3B. (17) Additionally, p53 actively participates in lineage commitment and cellular differentiation of stem cells through epigenetic regulation."
- In section 3, DBD abbreviation is used first time without mention of its full form.
-> The abbreviation ‘DBD’ was expanded to its full form upon its first appearance in the main manuscript.
- In section 4, authors change the method of writing and discuss one paper at a time explaining their findings and contribution. It would be better if this part is written as a coherent discussion instead. Authors could add their opinions, predictions, and speculations in a better way if this part is like a discussion.
-> I aimed to provide future readers of my article with an understanding of how these prediction models were developed. Therefore, I explained each model individually and added a detailed description of RFS model, along with the authors’ opinions and predictions, immediately after explaining each model as follows.
“These two models were established for predicting TP53-mutated head and neck cancers, and their consistent application to patients with AML.”
“In this study, the authors applied three different prediction models (disruptive vs. non-disruptive, EAp53, and RFS) to a cohort of patients with TP53-mutated AML. Variables of disruption (yes/no) and EAp53 (<75 or ≥ 75) did not prove to be significant predictive factors for OS and EFS. However, RFS (≤ -0.135 or >-0.135) emerged as a significant predictive factor for OS (12.9 months for patients with RFS ≤ -0.135 and 5.5 months for patients with RFS >-0.135, p = 0.017) and EFS (7.3 months for patients with RFS ≤ -0.135 and 5.2 months for patients with RFS >-0.135, p = 0.033) in multivariate analysis. Although this last prediction model can be applied to patients with TP53-mutated AML, it was not derived from a tissue-specific approach targeting TP53-mutated cell types, but rather from a mutation-specific approach. Consequently, we anticipate future prediction models based on findings from TP53 mutations in AML cells.”
- In section 5 again, there could be more addition to the discussion rather than mere stating facts.
-> I aimed to guide readers in thinking and reasoning how leukemogenesis occurs in TP53-mutated AML through a sequential arrangement of facts. To facilitate this, I added further sentences for discussion, as follows.
“Although TP53 is one of the genes found in age-related clonal hematopoiesis, (41) it appears that TP53 exerts a more deterministic influence compared with other genes such as DNMT3A, TET2, and ASXL1 when it comes to leukemic transformation.”
“It is worth noting that terminally differentiated hematopoietic cells like granulocytes may have limited opportunities to preserve acquired somatic mutations due to their relatively short lifespan.”
“The leukemogenesis mechanism in TP53-mutated AML might involve multiple hits accelerated by a deficient DNA repair mechanism resulting from TP53 mutation, rather than epigenetic dysregulation due to DTA mutations.”
- In the last paragraph of section5, there are no references. Are these all authors’ original ideas?
-> Several references have been added in the last paragraph of section 5
- In section 6, authors discuss reports of 83% of TP53 mutation frequency in AML patients without discussing it further. It might be good to discuss if this is higher frequency than predicted? If so, why would it be higher? What roles of TP53 would be important for such a phenotype?
-> Yes, higher frequency of TP53 mutation in secondary or therapy-related MDS/ALM with complex karyotype is well documented. This is attributed to previous chemotherapy or gamma-irradiation therapy potentially enriching resistant clones with TP53 mutation and leading chromosomal abnormalities due to their characteristics of chemotherapy resistance. However, the specific roles that TP53 plays in the development of complex karyotypes remain unclear.
- Right after talking about mutation frequency and complex karyotype, authors jump to chromothrypsis and telomere content very abruptly. Some introduction about chromothrypsis particularly why chromathrypsis may be connected to TP53 mutations, complex karyotypes etc. might be useful for readers. Similarly, before jumping to telomeres, why TP53 may be important for telomere maintenance could add some context for the readers.
-> Thank you for your valuable comments. I have added the following paragraph to address the high frequency of TP53 mutations, connecting it to the discussion in the paragraph about mutation frequency and its relation to chromothripsis and telomere.
“The high incidence of TP53-mutation in MDS/AML is closely linked to the presence of a complex karyotype. Both of these abnormalities are associated with prior therapeutic interventions that induce DNA damage. However, it remains unclear how TP53 mutation contributes to the development of a complex karyotype in hematopoietic stem and progenitor cells.”
- Paragraph about single-institute trail of decitabine could be re-written with better flow and coherence.
-> The term of “single-institution trial of decitabine” seemed inappropriate as you mentioned. I have revised it to “A cornerstone trial of decitabine”.
- Instead of writing ‘this result was also confirmed by a retrospective study’ authors could discuss the two studies together and state merits or demerits of one study over the other.
-> According to your suggestions, I have revised the paragraph to provide readers with an understanding of the strengths and weakness of two studies.
“Two studies investigated the benefits of venetoclax in patients with TP53-mutated AML. In the first study, a prospective phase 2 trial, investigators tested 10-day decitabine plus venetoclax 400mg daily followed by a 5-day decitabine plus venetoclax after achieving remission in newly diagnosed AML. The overall response rate and complete remission rate were 89% and 77% in WT-TP53 AML, whereas they were 66% and 57% in TP53-mutated AML. Survival outcomes were less favorable than response rate with a 60-day mortality of 26% vs. 4% and overall survival of 5.2 and 19.4 months in TP53-mutated and WT-TP53 AML, respectively. [68] In the second study, a retrospective analysis aimed to compare venetoclax-based and non-venetoclax-based regimens in TP53-mutated AML. In this study, investigators could not identify a significant difference between the venetoclax-based regimen and the non-venetoclax-based regimen in terms of OS (median OS, 6.6 vs. 5.7 months, p = 0.4) and relapse-free survival (median relapse-free survival 4.7 vs. 3.5 months, p = 0.43). [88] The first study was prospective and had a homogeneous population with the same treatment protocol, but it lacked a comparator group without venetoclax. On the other hand, the second study compared TP53-mutated and TP53-wild type AML, despite the treatment being heterogeneous and retrospective.”
- The discussion about ongoing clinical trails for TP53 mutated AML patients is well-written and the idea of having both table and discussion is well suited for this review’s purpose.
-> Thank you for your kind words.
Reviewer 3 Report
The manuscript titled "TP53 Mutation in Acute Myeloid Leukemia: An old foe revisited." provides valuable insights into the challenges associated with TP53 mutations in AML. The authors discuss the impact of TP53 mutations on treatment response and prognosis, as well as ongoing research efforts to develop effective therapies for TP53-mutated AML. Overall, the manuscript is well-written and provides a comprehensive overview of the topic.
Suggestions:
· Organization and structure: The article is divided into logical subsections that are in line with the flow of the article. However, the author has opted for multiple 1 or 2 line paragraphs within these subsections across the manuscript. I believe that is not necessary and disrupts the flow of the sub section being addressed. Eg. Line 112
· Line 77. First use of abbreviation DBD: Maybe addition of full form will be easier for readers to follow. "DNA binding domain (DBD)"
· Section 4. Scoring systems of TP53 mutations for clinical application:
Line 131-132 being the only study focused on AML that is cited for this sections, I believe elaborating the questions being asked in that study, methods adopted, data and findings of OS and EFS association could be helpful to the readers.
· Line 136-138: The text, along with preceding lines, implies that therapies cause mutational events leading to TP53 mutations emergence. Reference 34 is cited, which contradicts this idea. Clarification is needed in the text to align with the cited source.
Authors summary of the paper: “A longitudinal study tracking the evolution of mutation revealed that TP53 mutation is the first mutational event in chemotherapy or radiation therapy-induced MDS/AML”
Quote from the cited paper: “These data suggest that cytotoxic therapy does not directly induce TP53 mutations. Rather, they support a model in which rare HSPCs carrying age-related TP53 mutations are resistant to chemotherapy and expand preferentially after treatment.”
Author Response
Organization and structure: The article is divided into logical subsections that are in line with the flow of the article. However, the author has opted for multiple 1 or 2 line paragraphs within these subsections across the manuscript. I believe that is not necessary and disrupts the flow of the sub section being addressed. Eg. Line 112
-> Thank you for your comments. As you mentioned, there were several instances in my original manuscript where sentences were written as separate lines. I have now removed those extra spaces between separate sentences in most parts.
- Line 77. First use of abbreviation DBD: Maybe addition of full form will be easier for readers to follow. "DNA binding domain (DBD)"
-> The abbreviation ‘DBD’ was expanded to its full form upon its first appearance in the main manuscript.
- Section 4. Scoring systems of TP53 mutations for clinical application:
Line 131-132 being the only study focused on AML that is cited for this sections, I believe elaborating the questions being asked in that study, methods adopted, data and findings of OS and EFS association could be helpful to the readers.
-> I have incorporated the following paragraph based on your feedback. My article has now become more informative. Thank you for your assistance.
"In this study, authors applied three different prediction models (disruptive vs. non-disruptive, EAp53, and RFS) to a cohort of patients with TP53-mutated AML. Variables of disruption (yes/no) and EAp53 (<75 or ≥ 75) did not prove to be significant predictive factors for OS and EFS. However, RFS (≤ -0.135 or >-0.135) emerged as a significant predictive factor for OS (12.9 months for patients with RFS ≤ -0.135 and 5.5 months for patients with RFS >-0.135, p = 0.017) and EFS (7.3 months for patients with RFS ≤ -0.135 and 5.2 months for patients with RFS >-0.135, p = 0.033) in multivariate analysis."
- Line 136-138: The text, along with preceding lines, implies that therapies cause mutational events leading to TP53 mutations emergence. Reference 34 is cited, which contradicts this idea. Clarification is needed in the text to align with the cited source.
Authors summary of the paper: “A longitudinal study tracking the evolution of mutation revealed that TP53 mutation is the first mutational event in chemotherapy or radiation therapy-induced MDS/AML”
Quote from the cited paper: “These data suggest that cytotoxic therapy does not directly induce TP53 mutations. Rather, they support a model in which rare HSPCs carrying age-related TP53 mutations are resistant to chemotherapy and expand preferentially after treatment.”
-> Thank you for highlighting the crucial part I almost missed. I’ve revised the sentence as follows.
“A longitudinal study tracking the evolution of mutation has demonstrated that TP53 mutations represent the primary mutational event in chemotherapy or radiation therapy-induced MDS/AML. However, intriguingly, pre-existing TP53 mutations exhibit resistance to chemotherapy or irradiation and selectively expand after treatment. It is not the case that cytotoxic stress does not directly induce TP53 mutations. (39)”
Reviewer 4 Report
I have a few suggestions to enhance the clarity, organization, and overall impact of the abstract.
-The abstract lacks a clear structure with distinct sections. I recommend organizing the abstract into sections such as "Introduction," "TP53 Mutation in AML," "Clinical Implications," and "Conclusion." This will help readers navigate through the information more effectively.
-The introduction could benefit from providing a brief context about the significance of TP53 mutations in cancer and AML before delving into specific details. This would help set the stage for readers and establish the importance of the review's focus.
- I encourage the inclusion of a clear statement that explicitly outlines the objective or aim of the review paper. This could be phrased as summarizing the current understanding of TP53 mutations in AML and their clinical implications.
- To improve readability, consider breaking down complex sentences into smaller, clearer statements. This approach would ensure that the information is presented in a straightforward manner and is accessible to a wider audience.
- The conclusion could be strengthened to provide a more impactful summary of the key takeaways, and to emphasize the urgency of further research and potential breakthroughs in the treatment of TP53-mutated AML.
Some sentences are convoluted and could be rewritten for clarity.
Author Response
-> Thank you for your insightful feedback. I have made significant revisions to the abstract based on your valuable comments. The abstract is now more organized and improved thanks to your input.
Round 2
Reviewer 2 Report
Comments:
The manuscript has improved significantly and all of my previous comments are satisfactorily addressed in this revised version. I have just two suggestions:
Line 87 fundamental characteristic of forming a p53 tetramer through OD: explain OD beforehand as readers may not get the abbreviation
Line 122 CKDN1A gene: Isn't it the CDKN1A gene (gene for p21, right?)
Author Response
-> Thank you for kind words.
OD was changed to ”tetramerization domain” for consistency throughout the article. Regarding the gene name, ‘CDKN1A’ is indeed correct. Thank you for correcting the mistake I almost missed.
Reviewer 4 Report
The manuscript has shown significant improvement; nevertheless, there are still some typos present in the text. Additionally, it might be beneficial to replace the current figure with a conceptual diagram illustrating the function of p53.
For both mice and humans, gene and protein names are typically written in italics, with the first letter capitalized, and all subsequent letters in lowercase. If the gene or protein name is an acronym, it should be written in all uppercase letters. Here's an example using the p53 gene:
Gene:
Human: TP53
Mouse: Trp53
Protein Name:
Human: p53
Mouse: p53
Author Response
-> Thank you for kind words. I carefully read through the article and corrected typos. Many points where TP53 was written in non-italics (esp. in the Table) were found and revised. Some other typos such as CKDN1A and OD were also corrected.
According to your opinion, I added a figure 2 showing the diverse function of p53.
